# Potential of Producing Compost from Source-Separated Municipal Organic Waste (A Case Study in Shiraz, Iran)

**Haniyeh Jalalipour [1],\*** , **Neematollah Jaafarzadeh [2]**, **Gert Morscheck [1]**, **Satyanarayana Narra [1,3]** and **Michael Nelles [1,3]**

[1] Department of Waste and Resource Management, Rostock University, 18051 Rostock, Germany; gert.morscheck@uni-rostock.de (G.M.); satyanarayana.narra@uni-rostock.de (S.N.); michael.nelles@uni-rostock.de (M.N.)

[2] Toxicology Research Center, Ahvaz Jundishapur University of Medical Sciences, Ahvaz, Iran; Jaafarzadeh-n@ajums.ac.ir

[3] The German Centre for Biomass Research, 04347 Leipzig, Germany

\* Correspondence: haniyeh.jalalipour@uni-rostock.de

**Abstract:** Developing countries face serious environmental, social and economic challenges in managing different types of organic waste. Proper treatment strategies should be adopted by solid waste management systems in order to address these concerns. Among all of the treatment options for organic waste, composting is the most approved method as an effective strategy to divert solid waste from landfills. This experimental research aimed to examine the potential of producing compost from source-separated municipal organic waste in Shiraz, Iran. Market waste (fruits and vegetables) and garden waste (plant residues) were used as the raw input materials. They were subjected to the windrow pile composting method in an open site area. The process was monitored against several physical, chemical and biological parameters. In-situ measurements (temperature and moisture content) were carried out on a daily basis. Sampling and lab analyses were conducted over the period of the biological treatment. The final product was of acceptable moisture and nutrient levels, pH, Electrical Conductivity (EC), and Carbon/Nitrogen ratio. All of the analyzed compost samples had lower concentrations of heavy metals than the Iranian and German standards. Overall, the results obtained revealed that composting is a promising method for municipal organic waste treatment. The findings also imply the effectiveness of the source-separation collection method in the production of high-quality compost.

**Keywords:** municipal organic waste; windrow composting; solid waste management

## 1. Introduction

Worldwide, the development of urban areas, as well as the rapid growth of the population, has resulted in production of waste in an alarming ratio. Organic waste is the main fraction of the Municipal Solid Waste (MSW) produced in urban settlements. It is estimated that 2.6 million tons per day of municipal organic waste are generated globally [1]. Countries face serious environmental, social and economic challenges to manage this waste stream in an environmentally-sound manner [2,3].

The significant proportion of organic materials contributes to a high moisture content, ranging between 55% and 60%, with a thermal value expected to be about 7–8 MJ/kg [4]. This results in the generation of an excessive amount of leachate and landfill gas in dump/landfill sites, leading to serious environmental threats such as the contamination of the soil, and surface and underground water. In order to address these concerns, proper treatment strategies should be adopted by Solid Waste

Management (SWM) systems. Among all of the management options for organic waste, composting is the most approved method [5,6]. It is an effective strategy to divert solid waste from landfills and improve the heating value of feedstock in case of energy recovery [7].

Previous studies confirmed that composting can reduce the volume of organic materials by more than 30% [8], and can convert them into a hygienic and valuable product. The biological process is carried out through microbial activity. The activity of microbial community is affected by the temperature, pH, particle size, moisture content, aeration, and electrical conductivity of the organic waste [9]. $CO_2$, $NH_3$ and $H_2O$ are the by-products of the process, which lead to the recovery of mineral nutrients (nitrogen (N), phosphorus (P) and potassium (K)). The final product, rich in humus, can be utilized for sustainable agricultural purposes. Compost also has the potential to be used for the bioremediation of soil polluted with heavy metals, as it can immobilize them in soil matrix [10].

The quality of the available municipal organic waste significantly depends on the collection method used. The source-separation of the raw organic materials plays a pivoting role in securing the quality of final the product. In fact, the latter is considered to be a major obstacle in developing countries. Iran, as one such country, lacks proper solid waste management systems [11]. The enacted legislation does not effectively address the segregation of different types of solid waste. Only around 7% of the total MSW generated is collected separately as dry recyclables by municipalities. What remains is collected as mixed waste, including more than 70% organic materials [12]. The mixed household waste stream is well-known for containing hazardous and toxic waste (e.g., batteries and paints). The existence of such impurities makes the proper implementation of the composting process questionable. The final product may contain heavy metals and/or foreign materials (e.g., glass).

However, recycling municipal organic waste by the composting method, and the utilization of organic residuals to produce soil conditioners has grabbed the local authorities' attention for many years. The first compost facility in Iran was established in Isfahan in 1969, alongside with a Material Recovery Facility (MRF). Today, it is estimated that 42 MRFs are in operation in the country, and 18% of all of the MSW brought to them is subjected to the composting process [12]. However, these facilities are faced with many operational problems to produce a final product of high quality. Their failure mainly results from the improper raw organic materials of mixed waste origin containing high levels of impurities.

In Iran, the MSW segregation system must be adapted stepwise. Launching the source-separation programs for household waste stream is more challenging, since it highly depends on the behavior of the waste generators. Therefore, capacity-building programs to raise public awareness are a pre-requisite for the development of a sustainable SWM system. Furthermore, the fees collected from households for MSW activities are only based on family size, whereas, in case of businesses, the amount of waste generated is determinative. Daily markets, restaurants, and coffee shops, etc., are obliged to pay more for receiving a day-to-day collection service. This could be used as a mechanism for the effective implementation of SWM. For example, in Shiraz, the fifth most populated city of the country, around 30% of the daily markets that generate > 5 kg/d organic waste are given two storage bins with different colors for wet/dry waste segregation. The organic waste collected is subjected to vermicomposting at the small scale. This treatment technique requires high amounts of fresh water. The shortage of water is a problem, especially in the southern parts of the country, including Shiraz.

This study aims to adopt an effective waste treatment model to convert source-separated organic fractions into value-added products under the existing SWM system in Iran. The main objective of this work is to practice windrow composting for segregated markets and garden wastes as a viable method in mega cities with established compost facilities. The feasibility of the suggested system is mainly due to its low initial investment, simple technology, and routine monitoring. To this end, this research was carried out to examine the potential of producing compost from source-separated municipal organic waste in Shiraz, Iran.

The experiment explores the measurable changes of the organic material—including physical, chemical, and biological properties—throughout the composting process. The research examines the

quality of final product attained from the practice of the state-of-the-art technology for the windrow composting process under arid and semi-arid weather conditions. It was set out to assess the potential of compost produced for agricultural purposes in the study area, and the effects of applying organic amendments to the soil. It is hoped that these findings could be of assistance to decision-makers in the field of SWM in Iran.

## 2. Materials and Methods

### 2.1. Experimental Site

The study was conducted in an established composting plant, located at the Barmshor landfill site, 18 km south east of Shiraz, the capital of the Fars province, Iran. The province is located in the southwest of the country, and is identified by different agricultural activity. Shiraz has a semi-arid climate condition with limited fresh water resources. The total population of Shiraz is around 1.6 million inhabitants, on a surface area covering 217 km$^2$ [13].

The amount of MSW produced per capita in Shiraz is assessed to be 0.645 kg per day. The main ratio of MSW is collected as mixed and low fractions of dry waste, which are segregated at source. The informal sector affects the MSW stream significantly, and it is estimated that half of the dry valuable waste is collected by them. Half of the collected mixed waste is subjected to material recovery, and the remaining half is landfilled directly (Figure 1). The disposal method is sanitary landfilling, and the landfill's trenches are equipped with a Landfill Gas (LFG) collection system. LFG is used for electricity production at the biogas power plant.

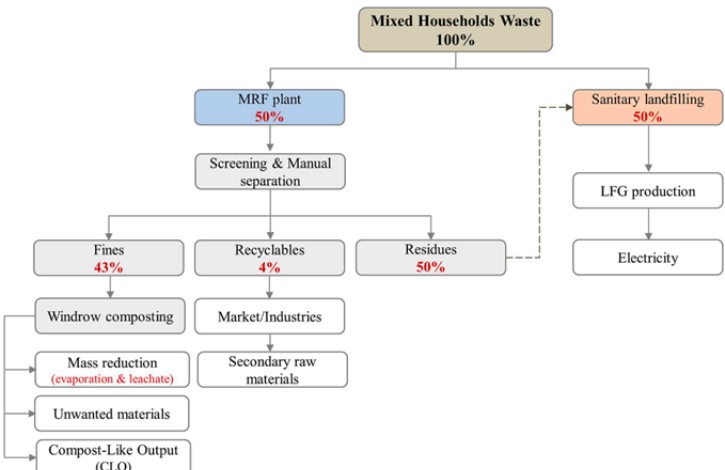

**Figure 1.** Material flow of municipal solid waste in Shiraz.

In the case of MRF, around 4% of the main input is separated manually. Recyclables are used as secondary raw materials in the recycling industry. The fine fraction (<70 mm) of about 43% is treated using windrow composting. The coarse fraction (>70 mm) is sent to the landfill.

### 2.2. Raw Materials

Different available types of organic waste, fruits and vegetables (market waste), and plant residues (garden waste) were used as the composting raw input material. The market waste included inorganic materials such as plastic bags. They were sorted manually. The garden waste, which was mainly comprised of dry leaves and clippings, was subjected to screening. Wood particles were shredded to wood chips. The characteristics of the initial raw materials are presented in Table 1.

**Table 1.** The characteristics of the initial raw materials used in the composting.

| Parameter | Fruits and Vegetables | Plant Residues |
|---|---|---|
| Physical properties | | |
| Bulk density (kg/m$^3$) | 628.69 | 314.2 |
| Moisture Content MC (%) | 80 | 7 |
| Chemical properties | | |
| Organic matter (%) | 54.30 | 55.00 |
| Total organic Carbon (%) | 30.17 | 30.56 |
| Total Nitrogen (%) | 1.4 | 1.1 |
| C:N Ratio (w/w) | 24.55 | 27.78 |
| pH | 7.10 | 7.7 |
| EC (dS/cm) | 2.72 | 2.98 |
| Total P (%) | 0.21 | 0.24 |
| Total K (%) | 0.6 | 1.4 |

*2.3. Methodology*

In order to conduct the experiment, source-separated market wastes were received over a period of 8 days in late February, 2019. The proper quantity of fresh materials was received to shape a windrow pile to be aerated using a machinery turner. Since the available market wastes (fruits and vegetables) did not meet the desired C/N ratio, materials rich in C (plants residues) were added as bulking agents to provide the required C/N ratio necessary for effective decomposition [14]. The mixture was chosen with respect to the needed C/N ratio of 25–30 for the beginning of the composting process [15]. The appropriate mixing of carbon and nitrogen benefit the procedure by providing an adequate source of food and energy for the microbial community. Consequently, around 50 tons of fruits and vegetables, and 17 tons of plant residues were combined in a 1:3 ratio in the experimental pile (Table 2).

**Table 2.** Compost pile ingredients.

| Raw Input Material | Initial Weight | Pile Dimension (W, H, L) |
|---|---|---|
| Fruits & vegetables | 50.40 tons | 3.5 m, 1.2 m, 20 m |
| Dry leaves | 16.80 tons | |

After blending the raw input materials thoroughly, they were aligned in a long windrow pile using a front-end loader. The dimension of the windrow has high importance in the maintenance of the composting process; height ranges of 1–2 m are crucial for heating in the pile, and the arched surface shape with a slow slope allows a higher absorption of irrigation/ rainfall water. The height and width of the pile is usually set according to the operating range of the machinery turner. The types of the raw materials and the mixture ratio affect the duration of the process and the properties of the final products.

The composting method implemented in the Shiraz plant is based on the principles of windrow technology in an open site area. Once the windrow pile has an adequate C/N ratio and bulk density, the aerobic composting process is dependent on providing moisture and oxygen as the necessary requirements for the activity of the microbial community. The in-situ monitoring of the composting process is a must for the production of a secured final product. The turning schedule provides the microbial population with oxygen, and adjusts the moisture content and temperature to an optimum level, as well as speeding up the stabilization of the organic materials [16].

Over the period of the composting process (10 weeks), the temperature and moisture content were monitored on a daily basis. The temperature was measured using a digital thermometer (model TESTO 925, Alton Hampshire, United Kingdom) in at least five points along the windrow pile, and in three depths (30 cm from top, middle, and bottom). The organic wastes used in this research work contained

adequate moisture to start the decomposition process. A Portal Moisture-meter (model REOTEMP, San Diego, CA, United States) was used to determine the moisture content inside the pile. The moisture was kept close to 50% for the first 4–6 weeks.

A specialized windrow turner (komptech topturn x53, Hirtenberg, Austria) was used to turn the piles. Turning schedule was adopted based on the temperature and moisture level. The schedule included three turnings in the first and second weeks, and two turnings in third to fifth weeks; from the sixth week onwards, the pile was turned once a week if the microbial activity continued. In order to monitor the physical and chemical operating parameters in different phases of the composting process, samples were taken. The sampling intervals were varied in order to determine the effects of different parameters. Since the moisture content significantly affects the biological activities, it was measured weekly. The EC, pH, total N, carbon and density were measured every other week. This time intervals was chosen because taking more samples (every week) could inhibit the composting process. Phosphorus (P) and potassium (K) have lower decomposition rates; therefore, their evolutions were measured at the beginning and at the end of the process. The parameters that are more crucial to agricultural applications (heavy metals, pathogens, phytotoxicity) were measured only at the end of the process.

Every time, at least five points along the windrow piles were chosen for sampling. A section was created by a mini loader in each point in such a way that the materials from the top, middle and bottom part of the pile could be picked and mixed thoroughly. A representative sample of >2 kg was taken using the three times quartering method, placed in a plastic container with a lid, and immediately sent to the lab. The laboratory analysis was carried out on three subsamples at the Shiraz municipal solid waste organization laboratory and the central laboratory of the Isfahan University of Technology (for the heavy metal content).

The moisture content was measured following equation 1; the representative sample was placed in a ceramic crucible and held in an oven set at 105 °C for at least 24 h, or until no difference in the weight measurement occurred [17]. The moisture was measured three times, and the average was then taken.

$$\textbf{Moisture Content} \, (\%) = \frac{(\textit{Wet sample weight} - \textit{Dry sample weight})}{(\textit{Wet sample weight})} \times 100\% \qquad (1)$$

The ash content was estimated by calculating the residual mass after heating. In total, 10 g of the prepared samples was placed in a ceramic crucible and subjected to muffle furnace heating up to 550 ± 10 °C for 6 h under strict time-controlled conditions, sample mass, and equipment specifications. The total amount of the organic carbon was calculated from the ash content, using Equation (2) [18].

$$\textbf{Total organic carbon} = \frac{\textit{Volatile solid} \, (100)}{1.8} = \frac{100 - \textit{Ash} \, (\%)}{1.8} \qquad (2)$$

The EC and pH were evaluated in 1:10 w/v sample: water extract using an EC/pH meter with a glass electrode [19]. The Total Kjeldahl Nitrogen (TKN) was analyzed using the regular Kjeldahl Method using a Gerhardt Vapodest 30S Analyser Unit (Königswinter, Germany). The total phosphorus (P) was measured using the spectrophotometry method (WTW Model photo Lab 6600 uv-vis, Weilheim, Germany). For the heavy metals, an Inductively Coupled Plasma-Spectrometer (Perkin Elmer, 7300 DV, Shelton, CT, United States) was used. The total potassium (K) was measured using flame photometry [17].

The Germination Index (GI) was measured using lab experiment [20]. About 10 mm of 1:10 compost extract was added to ten Garden Cress placed in petri dishes, with three replications. The petri

dishes were placed in the dark for 24–48 h, at room temperature. Water was used as the witness, and the germination index (GI) was calculated using Equation (3).

$$\mathbf{GI}\,(\%) = \frac{(Number\ of\ germinated\ seed\ \times\ root\ length)\ in\ compost\ extract}{(Number\ of\ germinated\ seed\ \times\ root\ length)in\ distilled\ water} \times 100 \qquad (3)$$

## 3. Results

### 3.1. Physical Properties

#### 3.1.1. Temperature

The oxidation of carbon sources by microbial activities produces abundant energy in form of heat during the decomposition process [21,22]. The temperature fluctuation is considered a proper indicator for an efficient composting process [23]. In this study, the temperature was monitored daily (Figure 2). The temperature rose to the thermophilic phase (>50 °C) in the first week, and reached to its highest level by the seventh day. This displays accurate initial conditions in the experimental pile. The rapid change in the temperature profile during the first week also indicates an extreme reduction in the organic matter [23]. During the active decomposing phase (the first three weeks), more temperature fluctuation was observed, which consequently required more turnings to maintain the process in the sanitation phase. The requirement for sanitation; the elimination of pathogens, insects, larvae and weed seeds by reaching the temperature ≥60 °C for more than 2 weeks was fulfilled [24,25]. The composting process of different organic materials took place over 10 weeks [26,27]. The lack of rises in temperature from the eighth week onwards confirmed the decreasing microbial activities. The temperature nearly reached the ambient from week 10, which proves the curing phase.

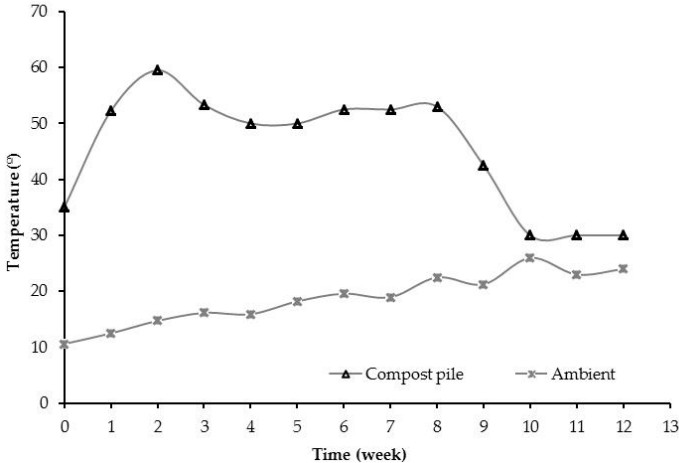

**Figure 2.** Temperature profile during the composting experiment.

#### 3.1.2. Moisture

Moisture is an essential factor in the maintenance of any metabolic process. Microbial metabolism requires an aqueous medium to gain nutrients and energy from chemical reactions. The moisture content impacts the process in terms of the oxygen uptake rate, the free air space and the temperature [28]. According to [22], the optimum moisture for an effective decomposition significantly depends on the waste nature. However, a moisture content between 40–60% during composting is crucial for microbial activity [29].

During the initial phase of this experiment, there was no leachate production. It was assumed that water was only lost through evaporation. This was due to the fact that the composting process was carried out under optimal conditions, with a well-blended pile structure (mixing ratio). Moreover,

the mild weather in winter during the composting process period inhibited the generation of leachate. The active decomposition rate in the initial weeks lead to the reduction of the moisture inside the pile. The moisture content was kept close to 50% throughout the active decomposition process until the tenth week (Figure 3). In addition, 163 mm rainfall during the composting experiment (February to June 2019) provided the pile adequate moisture content.

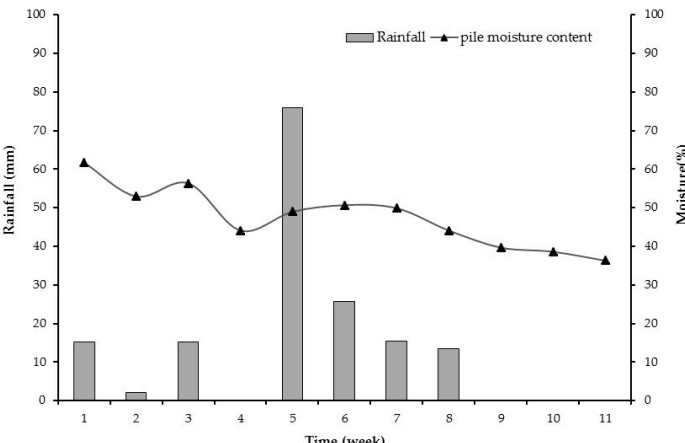

**Figure 3.** Rainfall frequency and moisture profile during the composting experiment.

The proper moisture content, adequate porosity, and accurate aeration sped up the organic matter's aerobic degradation. The evaporation rate is higher in semi-arid areas and, as a result, the moisture content decreased with a steep slope in the curing phase. At the end of the composting process, the moisture content value was around 36%. This is a rather remarkable outcome. It gives insight into the processing and storage conditions of the final product, which is a significant matter of concern in semi- arid regions.

### 3.1.3. Bulk Density and the Weight of the Pile

During the composting, aerobic microorganisms break down the organic materials into stabilized substances through several chemical reactions: mineralization, ammonification, and denitrification, etc. The byproducts of biological process are heat, carbon dioxide, ammonia, and humus-containing products of lower weight and volume [14,30]. The mass and volume reduction during the composting of several raw materials is a key factor in the design and operation of composting facilities. In this study, the mass and volume of the organic materials was reduced by approximately 27% and 21%, respectively, over the period of the composting process. The reduction values meet the results of [31], which reported that the mass loss during composting of six different feedstocks ranged from 11.5% to 31.4% of the initial weight, and 18.5% to 57.9% loss of the initial volume.

The bulk density is an indicator for the mass of the material within a specified volume [32]. It defines the quantity of the compost that can be placed at a certain site, as well as the machinery size for its transportation [33]. The waste within the composting pile achieves a higher bulk density by the end of the decomposition process [34]. The stabilization of organic materials leads to the enhancing of the ash content in such a manner that less volatile solids remain. Figure 4 displays the scatter diagram of the relationship between the bulk density and the volatile solids during the composting trial. As it shows, a significant negative correlation was found.

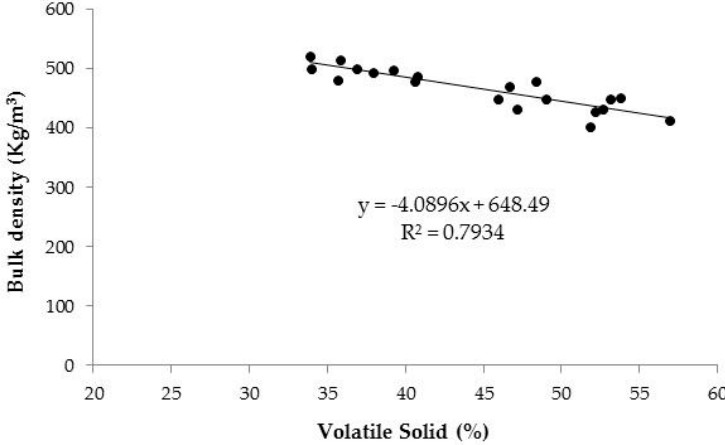

**Figure 4.** The relationship between the bulk density and volatile solids.

Overall, a remarkable increase in the bulk density of about 20.74% was observed in the experimental pile, while the volatile solids had a 34.75% reduction. These data reproduce the results from the study carried out by [35], who revealed that composts with a moisture content of 35–55% generally have a bulk density of 500–700 kg/m$^3$.

## 3.2. Chemical Properties

### 3.2.1. pH

The microbial activity in the soil and the availability of different nutrients to the plant's roots are considerably affected by the pH of the growth medium. The optimal pH value for microbial activity is suggested to be around neutral ranges of 6.5–7.5 [36]. The composting process began with a neutral range in the experimental pile (Figure 5). A lower pH value may result from the formation of fatty acids during the early stages of the composting process. Subsequently, during the mesophilic phase, the mineralization of nitrogenous compounds increases the pH value through the formation of ammonia ($NH_3$) and ammonium ($NH_4^+$) [37]. In the curing phase, the oxidation process transforms ammonium ($NH_4^+$) to nitrate ($NO_3^-$), which is defined as the nitrification process [38]. The pH value of the final product usually reaches a constant level ranging from 7.2 to 8.3, depending on the composting feedstocks [39].

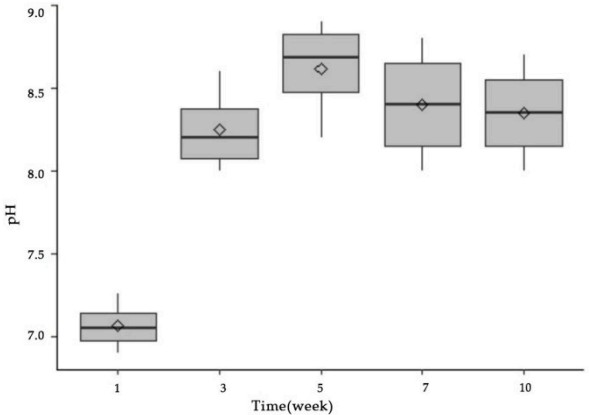

**Figure 5.** pH values of the composting experiments.

Looking at Figure 5, it can be seen that a pH value of 8 was reached in a short period. This resulted from the considerable breakdown rate, supported by the high turning frequency during the initial weeks of the process. The final product indicated a reasonable range of pH for finished compost [35].

### 3.2.2. Electric Conductivity (EC)

Electrical conductivity (EC) is usually measured in soil and compost in order to estimate the salinity of the growth media [40]. EC is an important chemical property for the end users of compost products due to the toxic effects that it may have on plant growth [41]. The addition of compost with high EC to the soil increases the salt accumulation in the root zone and inhibits water absorption by roots [40]. Crops can resist different ranges of salinity according to their species. The EC of the final product has a high importance in arid and semi-arid conditions due to the lower organic matter content and higher salinity level of soils in these areas.

The degradation of organic materials by the soluble salts released by the microbial activity during the composting process consequently increases the EC value [33]. Over the composting period, the EC slowly increased to be around 3.18 dS/m, and this range remained more or less stable until the end of the process, resulting in a final EC of 3.36 dS/m after 10 weeks (Figure 6). Compared to the preferred range of EC values for the growth medium, 2–4 dS/m, the final product has a high potential to be used as a soil conditioner [34].

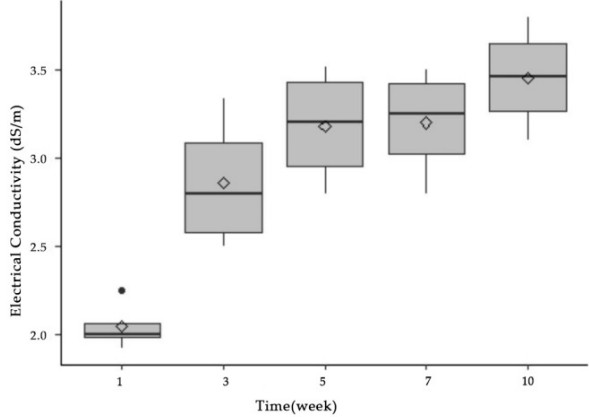

**Figure 6.** EC profile of the composting experiment.

### 3.2.3. C/N Ratio

The relative proportion of carbon and nitrogen is one of the major parameters in controlling the nutrient balance in the composting process [22]. Carbon acts mainly as a source of energy for the microorganisms. The population growth of bacteria is influenced by the nitrogen fraction, because their structure is mainly formed from proteins [38]. The ideal range for the stimulation of nitrogen immobilization or mineralization in the composting process falls in the initial substrate of about a 25 to 35 C/N ratio [42].

The decomposition process is slowed down by limited nitrogen sources, whereas the mixtures with a low carbon content lost extra nitrogen as ammonia during the turnovers [38]. In order to improve the decomposition process, it is essential that carbon is provided in easily-degradable forms. For example, materials of high cellulose content should be shredded prior to piling in order to become readily available for microorganism consumption. The microbial activity significantly decreases in a <20 C/N ratio. For that reason, the C/N ratio is mostly used as stability index, ranging from 15 to 20 for a finished final product.

Figure 7 presents the average value of the C/N ratio during the composting period. In this figure, there is a clear trend of a decreasing C/N ratio, with a significant reduction of about 36% by the end of the process. It is apparent from this data that the organic waste was degraded effectively. This was due to the ideal conditions in the experimental pile created by providing a proper source of carbon and nitrogen. As a result, the final product appeared to be stable, with an average C/N ratio of about 16.42.

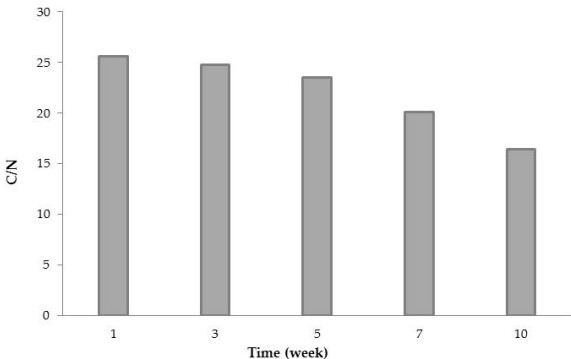

**Figure 7.** C/N ratio profile during the composting run.

### 3.2.4. Nitrogen (N), Potassium (K), Phosphor (P)

From an agricultural point of view, the fertilizers should provide the soil with essential nutrient elements to support the crops' development stages. Nitrogen (N), Potassium ($K_2O$), and phosphorus ($P_2O_5$) are the macro nutrient elements that significantly affect the plant growth. Usually, small parts of nutrients are mineralized by microbial activity over the period of the composting, resulting in the higher nutrient content of the final product [43]. In this research, the concentrations of the total N, $K_2O$ and $P_2O_5$ was measured to be 12%, 7%, and 33% higher in final product, respectively (Figure 8).

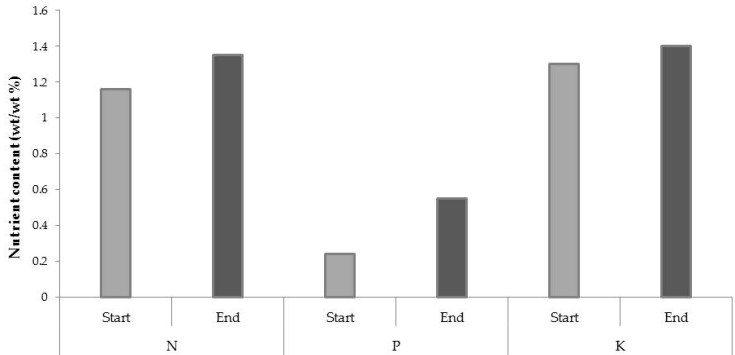

**Figure 8.** Nutrient content (wt/wt %) at the beginning and end of the composting experiment.

What stands out in Figure 8 is that the nutrient content of the compost produced depends significantly on the type of raw organic waste used [18]. Fruits, vegetables and plant residues have higher N and K (%), which can be seen clearly from these findings. One of the issues that emerge from these findings is that the concentration of phosphorus ($P_2O_5$) in the final product doesn't meet the values set up by Iran national compost standard for class 'A' compost (1–3.8%); therefore, nutrient adjustment using mineral fertilizer is needed prior to marketing.

### 3.2.5. Heavy Metals

The nature of the fresh organic waste and its collection method affects the heavy metal content of the compost produced [44]. Some authors reported composting as a method of heavy metal immobilization [45]. In fact, the biological process changes them into unavailable forms for plant uptake. Nevertheless, the mass and volume reduction of the waste during the decomposition process increase the concentration of heavy metals in the final product [46]. The utilization of compost with a high content of heavy metals can affect soil properties, reduce crop productivity, and contaminate the food chain in the long term [47]. The limitation on heavy metal content is therefore of particular concern in the application of the final product. The standard limit of heavy metals varies in different countries. Developing countries mostly have lower limitation levels, which allow them to use raw organics from mixed waste streams.

The results obtained from the analysis of the finished compost samples were compared with the Iranian and German standards in Table 3. The German standard has one of the strictest limitations on heavy metal content among developed countries. As can be seen from the table, the compost produced has lower heavy metal concentrations than the values set by the German standard. These results show the importance of waste segregation in order to ensure the quality of the final product.

**Table 3.** Heavy metal content compared with the Iranian and German standards.

| Parameter (mg/kg) | Value | Iran Standard | German Standard [48] | |
|---|---|---|---|---|
| | | | Class A | Class B |
| Pb | 27.35 | 200 | 150 | 100 |
| Cd | 0.04 | 10 | 1.5 | 1.0 |
| Cr | 8.32 | 150 | 100 | 70 |
| Cu | 56.89 | 650 | 100 | 70 |
| Ni | 15.34 | 120 | 50 | 35 |
| Hg | - | 5 | 1.0 | 0.7 |
| Zn | 95.26 | 1300 | 400 | 300 |
| Co | 0.4 | 25 | - | - |

### 3.2.6. Pathogen Content

The metabolic reactions under aerobic composting are mainly controlled by the temperature fluctuation [22]. This has considerable effects on the type, number and species of microorganisms involved in the biological process. Indeed, most pathogens are deactivated and destroyed at high temperatures [38].

Fertilizers can contaminate soil and plant roots in the case where they contain pathogens such as *E. coli* and *Salmonella* [49]. Several authors have reported that the thermal death point of 55–60°C for at least 24 h, alongside moisture of about 50% in the composting pile, is crucial for the removal of most types of pathogens [50,51]. Therefore, maintaining the composting process with favorable moisture and heat has high importance in securing the quality of final products.

The Iranian national standard defines limits only for the *E. coli* and *Salmonella* content of organic fertilizers. The results, presented in Table 4, indicate that the produced compost is pathogen free according to the local standards. As such, the final product could be safely applied for the cultivation of any crops.

**Table 4.** Pathogen content of the produced compost, compared with the Iranian standards.

| Type of Pathogen | Content (MPN/g) | Iran Standard |
|---|---|---|
| *E. coli* | <3 | $1 \times 10^3$ (MPN/g) |
| *Salmonella* | Absent | 3 (MPN/4 g) |

### 3.2.7. Phytotoxicity of the Final Product

A phytotoxicity test is used to describe the degree of the maturity of the final product. Maturity is defined as the state of being appropriate for agricultural purposes. The seed growth is influenced significantly by the existence of phytotoxic substances such as ammonia, organic acids, and heavy metals, etc. [52]. The organic acids produced during initial phase are almost degraded in the curing phase under optimum composting conditions. Salt and heavy metals are mineralized or immobilized [53], and nitrification contributes to the lowering of the $NH_4^+/NO_3$ [54]. During the curing phase, the microbial activity decreases and the pile temperature drops down to reach near the ambient temperature [55].

The Germination Indexes (GI) of sensitive seeds like Cress and Chinses radish are widely used to determine the toxicities of final products [56]. The compost produced by the experimental pile had an 80% GI in this research. Several studies reported the absence of phytotoxicity by GI ≥ 80% [57,58].

According to the Iranian standard, the final product should obtain a GI > 70% to count as a mature product. These findings indicate that the experimental pile was subjected to a complete curing phase, and the compost produced can be used for agricultural purposes safely.

## 4. Discussions

### 4.1. Scaling-Up the Collection of Source-Separated Organics

Shiraz city has two main organic waste streams: business units and households. The solid waste management organization is fully responsible for handling organic waste. The collection service is offered on a day-to-day basis to business units, and on every alternative day to the households.

Subjecting the household waste to a source-separated collection system is inapplicable under the existing waste management system. The absence of legal enforcement, the uncertain quality of the raw materials, the small quantity of daily waste generated, and the difficulty in storing it for a long time make this option out-of-scope.

Up to 30% of daily markets are covered with a source-separation collection system using five vehicles with different capacities, ranged from 2 to 5 tons. The amount of this organic waste stream accounts for 7 to 10 tons/day of fruits and vegetables. The remaining fractions are collected with the mixed waste stream. Besides this, around 800 tons of garden wastes are collected seasonally (from November to January).

In actuality, the existing collection system does not have the capability to cope with the amount of organic waste generated; the system has a shortage of collection vehicles and manpower. Up to 70% of market organics are therefore not covered through the segregated collection system. A plan should be developed that aims to upgrade the source-separation collection system. Consequently, the guaranteed available amount, freedom from impurities, and homogenous mixture of the market organic waste would create a high potential for the raw organic materials to end up in a composting facility.

In parallel with the separate collection of the whole amount of the organic waste generated from the markets, attention should also be paid to the quality of the collected segregated organics. This could be guaranteed by making the waste producers an active part of the desired system. Usually, market stores pay fees for each extra bin, which makes them unwilling to cooperate with the segregation. Carrying out incentive programs and awareness campaigns for them are the major tools to obtain the desired collaboration and contribution.

Another area that should be focused on is the catering business (restaurants, hotels and canteens). This sector is considered one of the main sources of the provision of rich nutrient-containing organics with a wide range of protein-based products. Overall, the guaranteed availability of such a mixture (fruits, vegetables, food waste, and garden waste) creates considerable potential for the production of compost with high quality.

### 4.2. The Positive Effect of Using Class Compost in Soil

One of the aims of this study was to examine the application of the final product for agricultural purposes. The finished compost with a desired quality has the potential to enhance the soil's physical, chemical and biological properties. The quality control parameters vary in different countries. They usually define agronomic aspects and the risk-free use of the final products [59]. National and international standards put great emphasis on the cleanliness of municipal solid waste compost regarding heavy metals, while fertility is a matter of marketability. For example, the assessment system developed by [11] evaluates the agronomical and environmental viability of compost produced from MSW, and, accordingly, its final application is distinguished. Following this approach, the compost produced in this research is of high fertility and risk-free. All of the final product samples appeared to be stable and were considered to be class 'A', making it suitable for the cultivation of any agricultural crops.

The application rate of fertilizer is estimated based on the type of crop, irrigation water (quality, availability), and the growth media properties. Compared to chemical fertilizers, the available form of

nutrients in the compost is lower, which raises the need for a higher rate of application to provide the crop with the required level of nutrients. In the study area, the application rate of 1.5 to 30 tons per hectare is typically recommended [60]. The results obtained from experimental analysis indicated that the final product contains N-K-P of about 1.35%, 1.4%, and 0.5%, respectively. Considering the semi-arid weather conditions, it is assumed that the compost has a 30% moisture content. Consequently, there is 9.45 Kg, 9.8 Kg, and 3.5Kg, respectively, of N-K-P in each ton of compost produced.

The application of compost leads to the improvement of soil fertility by increasing the organic matter, cation exchange capacity, and biological activity, as well as improving the water retention and temperature regulation. Several reports have shown that Soil Organic Carbon (SOC) can be increased by between 50 and 70 kg per ton of (dry mass basis) compost per hectare per year [61,62]. Ultimately, carbon sequestration contributes to prevent soil erosion and greenhouse gass emission. The benefit of compost application in the study area was evaluated by assuming a moderate range of SOC increase (50 Kg·ha$^{-1}$·y$^{-1}$·t$^{-1}$ dry mass) over periods of 10 and 20 years (Figure 9).

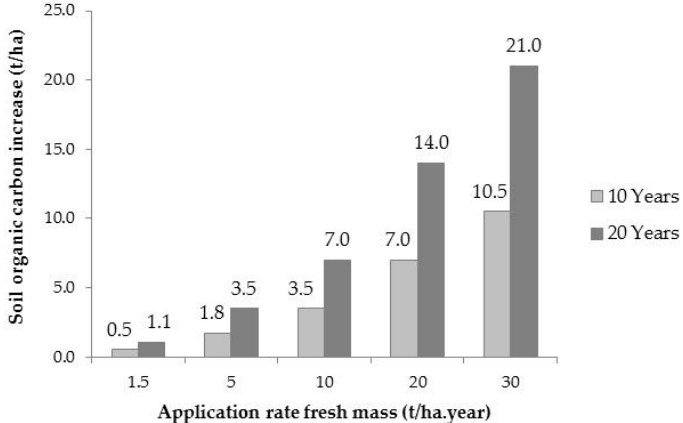

**Figure 9.** Estimated SOC increases under different compost application rates (fresh mass) over 10 and 20 years.

The figure above illustrates a direct relationship between the amount of SOC and the compost application rates. The highest amount of carbon sequestration therefore occurs when a higher rate of compost is applied. Over the periods of 10 and 20 years, SOC will be increased 10.5 t·ha$^{-1}$ and 21 t· ha$^{-1}$ with 30 tons per hectare compost application, which also has a considerable positive effect on the available soil water. However, these results may not be attained due to the lower application rate of compost on the basis of crop nutrient demands.

Another interesting finding is obtained by quantifying the amount of organic carbon sequestered in the soil into the equivalent carbon dioxide using the stoichiometric ratio of $CO_2$ (3.67). Assuming a moderate range of SOC increase, this means that the application of each ton of compost has the potential to sequester 128.45 kg carbon dioxide equivalent on a fresh mass basis (50 × 0.7 × 3.67). The carbon sequestrated in soil stated as carbon dioxide equivalents for different compost application rate is shown in Table 5, over periods of 5, 10, 15 and 20 years.

**Table 5.** Sequestered carbon in carbon dioxide equivalents.

| Compost Application Rate (tons/hectare) | Time (year) | | | |
|:---:|:---:|:---:|:---:|:---:|
| | **5** | **10** | **15** | **20** |
| 1.5 | 0.96 | 1.93 | 2.89 | 3.85 |
| 5 | 3.21 | 6.42 | 9.63 | 12.85 |
| 10 | 6.42 | 12.85 | 19.27 | 25.69 |
| 20 | 12.85 | 25.69 | 38.54 | 51.38 |
| 30 | 19.27 | 38.54 | 57.80 | 77.07 |

Carbon sequestration is significant at the application rate of 20 and 30 tons of compost per hectare. This would also have the effect of preventing soil erosion, which is in agreement with sustainable development goal target 15.3, i.e., combat desertification and restore degraded land and soil by 2030.

## 5. Conclusions

This project was undertaken in order to adopt a feasible method for the effective treatment of municipal organic waste in mega cities in Iran. Therefore, segregated municipal organic wastes were subjected to aerobic pile composting technology in an established compost plant in Shiraz, Iran. A windrow pile was prepared from source-separated market and garden waste. The raw organic materials were blended with 1:3 ratios in order to ensure the required initial C/N ratio for efficient microbial activity.

In-situ measurements were carried out on daily basis; the temperature and moisture were monitored and, accordingly, the pile was subjected to aeration. Sampling was carried out periodically; representative samples were taken every alternative week in order to monitor and ensure an accurate degradation process. Lab analyses were performed in equipped laboratories for several key parameters; moisture content, bulk density, pH, EC, Nitrogen, C/N, and nutrient contents. The final product was subjected to a strict evaluation. The finished compost was examined for pathogens, heavy metals and phytotoxicity, and the obtained results were compared with the values set by the national and German standards.

The final product of the experimental pile was of acceptable moisture, nutrients, pH, EC, and C/N ratio. All of the analyzed samples appeared to be pathogen-free. Concerning the heavy metals, all the compost samples analyzed had lower concentrations than those values set by the Iranian and German standards. The findings indicate that the state-of-art technology for the composting process was performed successfully under ideal conditions. The characteristics of the final product demonstrate a complete degradation of the organic waste within a relatively short period of time.

Indeed, after the successful implementation of this pilot project, the number of daily markets subjected to the source-separation program were increased. This was mainly due to the fact that, compared to vermicomposting (on average 3 months), this method could turn the organic materials, in 10 weeks, into a value-added product. Furthermore, it requires less labor work, and the process is less vulnerable to semi-arid weather conditions.

The findings of this study suggest that a wet/dry source-separation program can be effectively adopted for business units in Iran. Indeed, in the initial phase, among all of the municipal organic waste streams, the segregation and separate collection of market and garden waste are more viable under the existing SWM system. These findings are particularly relevant to decision makers in mega cities with established compost facilities.

**Author Contributions:** Conceptualization, H.J., S.N. and G.M.; methodology, H.J., S.N. and G.M.; investigation, H.J.; data curation, H.J.; writing—original draft preparation, H.J.; writing—review and editing, H.J., G.M. and S.N.; supervision, S.N., G.M. N.J. and M.N. All authors have read and agreed to the published version of the manuscript.

**Funding:** The research work was funded by the Solid Waste Management Organization (SWMO) of Shiraz municipality by providing all of the requirements needed to perform the study in terms of raw input materials, working areas, machineries, equipment, staff and laboratory analysis. The research was also funded by Deutsche Forschungsgemeinschaft and Universität Rostock within the funding programme "Open Access Publishing".

**Conflicts of Interest:** The authors declare no conflict of interest. The funders had no role in the design of the study; in the collection, analyses, or interpretation of data; in the writing of the manuscript, or in the decision to publish the results.

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
