# Peer review of "Potential of Producing Compost from Source-Separated Municipal Organic Waste (A Case Study in Shiraz, Iran)"

_sustainability, doi:10.3390/su12229704_

Round 1
Reviewer 1 Report
In study: "In Iran, more than 70% of collected MSW is organic materials..." ButI am missing from the study a proposal for a sorting method for separately collected waste. I think could be complemented by this study with this method.
Author Response
Response 1:
Source-separation of municipal solid waste was considered as a crucial point for producing high quality compost in Iran. Different strategies should be considered for main municipal organic waste streams; households and business units. The public (households) should be subjected to intensive capacity-building programs and awareness campaigns. The commercial sector is more flexible for establishment of waste segregation system due to the existence of effective legal framework. It was tried to indicate these point in introduction from line 72 to 82 (in final version) as followed:
“In Iran, MSW segregation system must be adapted stepwise. Launching the source-separation programs for household waste stream is more challenging, since it highly depends on the behaviour of waste generators. Therefore, capacity-building programs to raise public awareness are a pre-requisite for the development of a sustainable SWM system. Furthermore, the fees collected from households for MSW activities are only based on family size whereas in case of businesses, the amount of waste generated is determinative. Daily markets, restaurants, coffee shops and etc. are obliged to pay more for receiving a day-to-day collection service. This could be used as a mechanism for the effective implementation of SWM. For example, in Shiraz, the 5th populated city of the country, around 30% of the daily markets that generate >5 kg/d organic waste are given two storage bins with different colors for wet/dry waste segregation. The organic waste collected is subjected to vermicomposting in small scale or is added to the mixed waste composting piles.”

Reviewer 2 Report
The submitted paper entitled "Potential of Producing Compost from source-separated municipal organic waste (A case study in Shiraz, Iran)" intends to provide evidence of the advantage of composting as a promising method for municipal organic waste treatment. The paper deals with an important and practically valuable aspect. However, I suggest major revision of the manuscript.
Unfortunately, the overall writing should be polished in order to enhance clarity and readability. I think that the paper is written on such a manner that following the narration is quite difficult.
Comments:
Line 26" use "revealed" instead of "demonstrated"
Line 77: Please define what do you mean by "commercial sector"
Line 205-206: Please re-phrase.
Figure 3: There is a significant increase in moisture during the 3rd week, with almost no rainfall during weeks 2 and 3. Please clarify this observation.
Lines 240-241: How "strong relationship between bulk 240 density and volatile solid during composting trial" is concluded/confirmed
Line 244: please use the same units (kg/m3 or g/cm3) throughout the whole manuscript, to enhance readability
Line 251: The first sentence "pH value indicates ... (25 ËšC)" could be avoided. It's really fundamental knowledge
Line 255: The authors state that "The composting process began with acidic range in experimental pile (Figure 5)" but in figure 5 the pH at week 0 is nearly 7 !! I don't see how pH 7 could be characterized as acidic! Subsequently, they describe a mechanism for decreasing pH value but in figure 5 the pH of the experiment has an increasing trend. This is confusing.
Lines 309-313: "leachate resulting at lower content in final product"
"of organic materials slightly increased nutrient content of final product"
"The concentration of total N, K2O and P2O5 in final product, was measured 12%, 7% and 33% 312 higher, respectively"
In conclusion, I see the potential of the paper, but some aspects should be addressed.
Author Response
Point 1: Unfortunately, the overall writing should be polished in order to enhance clarity and readability. I think that the paper is written on such a manner that following the narration is quite difficult.
Response 1: The paper is revised in order to increase the readability and simplicity to follow the main object. In particular introduction was modified to address the story line of the research. Methodology, results and discussion are moderately corrected and conclusion is looked over to recap the message of the article.
Point 2: Line 26" use "revealed" instead of "demonstrated"
Response 2: demonstrated is used instead of revealed.
Point 3: Line 77: Please define what do you mean by "commercial sector"
Response 3: business units are used instead of commercial sector as one of the main generator of municipal organic waste in Iran. This includes mainly Daily markets, restaurants, café and etc.
Point 4: Line 205-206: Please re-phrase.
Response 4: the verb “should” is omitted and the sentence is re-phrased as followed:
“However, moisture content between 40 - 60% during composting is crucial for microbial activity [29].”
Point 5: Figure 3: There is a significant increase in moisture during the 3rd week, with almost no rainfall during weeks 2 and 3. Please clarify this observation.
Response 5: It was a mistake in converting date from Persian calendar to European. The rainfall in 4th weeks actually happened in 3th week and there was no rainfall in 4th weeks.
Point 6: Lines 240-241: How "strong relationship between bulk 240 density and volatile solid during composting trial" is concluded/confirmed.
Response 6: The paragraph regarding bulk density and related Figure (4) is revised. In old version, the averages of analyses in each sampling were used for regression between bulk density and volatile solid. The new figure shows all replications. The paragraph is modified to address the changes of volatile solid by increasing bulk density during composting process as followed:
“The stabilization of organic materials leads to enhancing the ash content in such a manner that less volatile solids are remained. Figure 4 displays the scatter diagram of the relationship between bulk density and volatile solid during composting trial. As it shows, a significant negative correlation was found. Overall, a remarkable increasing in the bulk density of about 20.74 % was observed in experimental pile while volatile solids had 34.75 % reduction.”
Point 7: Line 244: please use the same units (kg/m3 or g/cm3) throughout the whole manuscript, to enhance readability.
Response 7: kg/m3 was used in whole manuscript.
Point 8: Line 251: The first sentence "pH value indicates ... (25 ËšC)" could be avoided. It's really fundamental knowledge.
Response 8: This sentence was omitted “pH value indicates acidic or basic behaviour of an aqueous solution, based on calculation of hydrogen ions at room temperature (25 ËšC).”
Point 9: Line 255: The authors state that "The composting process began with acidic range in experimental pile (Figure 5)" but in figure 5 the pH at week 0 is nearly 7!! I don't see how pH 7 could be characterized as acidic! Subsequently, they describe a mechanism for decreasing pH value but in figure 5 the pH of the experiment has an increasing trend. This is confusing.
Response 9: This paragraph is modified as followed: “The composting process began with neutral range in experimental pile (Figure 5). Lower pH value may result from formation of fatty acids during early stages of composting process.”
Point 10: Lines 309-313: "leachate resulting at lower content in final product"
"Of organic materials slightly increased nutrient content of final product"
"The concentration of total N, K2O and P2O5 in final product, was measured 12%, 7% and 33% 312 higher, respectively"
Response 10: In this paragraph the following sentences are omitted: “Theoretically, some parts of nutrients are lost by evaporation and leachate resulting lower content of in final product. In practice, the reduction in mass and volume of organic materials increased nutrient content of final product slightly.”

Reviewer 3 Report
Dear authors,
the theme presented is current and of great interest. All the problems related to the potential of compost from source-separated municipal organic waste is clearly described in the introduction and discussion but the experimental data(parameters evaluated, statistics and tables) should be improved.
Although important biological parameters such as enzymatic activity were not considered, my greatest perplexity about the manuscript is related to the fact that the data lack replicates, the histograms do not show the standard deviation.
Further, some of minor revision are reported on the attached pdf. I hope that my suggestions could help to improve your manuscript.

Author Response
Point 1: the theme presented is current and of great interest. All the problems related to the potential of compost from source-separated municipal organic waste is clearly described in the introduction and discussion but the experimental data (parameters evaluated, statistics and tables) should be improved.
Response 1: The experimental data including figures and tables are modified. The replications are considered in drawing EC, pH and bulk density-volatile solids figures.
Point 2: Although important biological parameters such as enzymatic activity were not considered, my greatest perplexity about the manuscript is related to the fact that the data lack replicates, the histograms do not show the standard deviation.
Response 2: This study was designed and implemented as part of cooperation between Shiraz (Iran) Solid Waste Management Organization and department of waste and resource management of Rostock University (Germany). Therefore biological parameters such as enzymatic activity were not a matter of concern for both parties.
Concerning the lack of replicates, the histograms were drawn by using the average of values. Indeed, laboratories analysis was carried out on three subsamples from main sample. It is inserted at line 169-170 in methodology (final version).
The boxplot are used for EC, pH and bulk density-volatile solids are to indicate min, max, median and average. However, C/N ratio figure still shows the average value of all samples.
Point 3: Line 23: lower concentrations of? Which elements?
Response 3: All the compost samples analysed had lower concentrations of heavy metals than those values set by Iranian and German standards.
Point 4: Line 137: how long was the period?
Response 4: the period of composting process is mentioned as 10 weeks in line 146 (final version).
Point 5: Line 161 & 167: please, correct: Celsius degrees.
Response 5: Celsius degrees are corrected in line 161 and 167.
Point 6: Lines 230: Please, correct: Table 4 not 5. Anyway my opinion is that you could also to consider to remove this table, as the data were already reported in the text.
Response 6: The table and figure labels are revised in whole manuscript and table 4 is removed.
Point 7: Line 235: please, correct: bulk density.
Response 7: table 4 is removed from manuscript.
Point 8: Line 242: It should sound better…. These data reproduce results from the study carried out.
Response 8: This line: 255-256 (final versions) is re-phrased to “These data reproduce results from the study carried out by [35], who revealed that composts with moisture content of 35 - 55 % generally have bulk density of 500 - 700 kg/m3.”
Point 9: Line 307: Please correct and check all formula into the text.
Response 9: K2O, CO2, NH3, NH4+, NO3- and P2O5 are checked and corrected though the whole manuscript.
Point 10: Lines 317: figure 8 doesn't show clearly distinction between raw organic waste!
Response 10: The raw organic materials were all the same type. The paragraph is revised to convey the message better as followed: “What stands out in figure 8 is that nutrient content of compost produced depends significantly on the type of raw organic waste [18]. Fruits, vegetables and plant residues have higher N and K (%) which can be seen clearly from these findings”
Point 11: Lines 334: Please correct: Table 5 not table 6. Please remove mg/kg from the listed parameters and insert it on the first Parameter (mg/kg).
Response 11: The table and figure labels are revised in whole manuscript. Mg/kg from the listed parameters in table 3 (final version) is removed and inserted in the cell containing label.
Point 12: Lines 346: Please correct: Seems that period from Several authors to composting pile miss something….
Response 12: The sentence at line 349-351 is completed as followed: “Several authors have been reported that thermal death point of 55 - 60°C for at least 24 hours alongside with moisture of about 50 % in composting pile is crucial for removing most type of pathogens [51, 52].”
Point 13: Lines 348: Please, correct: Table 6.
Response 13: The table and figure labels are revised in whole manuscript.
Point 14: Line 351: Please correct: Title of 3.2.7 are the same of 3.2.6!.
Response 14: The headline title of 3.2.7 is corrected as “Phytotoxicity of final product”
Point 15: Line 352: Please correct: Phytotoxicity term and also in lines 361 and 460.
Response 15: Phytotoxicity is used in corrected form in section 3.2.7.
Point 16: Line 444: Table could be improved, it is not so clear, maybe you could to invert rows x columns. Please remove year from the list.
Response 16: Table 5 (final version) is modified. The columns and rows are converted and “year” is used only in the cell contains the label “time”.

Reviewer 4 Report
The work is well structured. However, I did not realize what the purpose of the work was, being that the thematic is already quite studied.
What is the main objective of the work?
The separation of the garbage was carried out directly at the source, an interesting idea on a small scale, but on a large scale is this possible? Is there feasibility for such a study?
The time of the composting test must be included in the methods, as well as the reasons why it led the authors to determine this working time.
As the authors can confirm an assay if only one assay was performed with each sample. Couldn't there have been errors regarding the operator or equipment?
Even without the production of leachate, contaminants, coliforms, protozoa and others that may be present in the material for composting, which may have generated soil pollution. As they ensured that there was no contamination.
With regard to pathogens, the analyzes were performed only at the end of the trial, how to ensure that the process was efficient for the treatment since you do not know how it is at the beginning?
In the conclusions between lines 468 and 474, were the results obtained in the study? Or were they based on bibliographic research? If it is based on the tests, which tests did you perform? How was it proven? A conclusion must be made based on the studies, other statements must be inserted in the text and duly referenced.
Author Response
Point 1: The work is well structured. However, I did not realize what the purpose of the work was; being that the thematic is already quite studied.
Response 1: Windrow composting started in Iran alongside with material recovery facility in 1969. The fine fraction of MRF out-put is used as raw input materials. Consequently, most previous studies on municipal composting in Iran were implemented using mixed waste stream. Despite some mega cities started to collect source-separated market organic waste after enactment of municipal solid waste national legislation in 2004, less attention are paid to windrow composting and mostly the segregated municipal organic waste are subjected to vermicomposting and/or animal food production in small scale. These treatment techniques require high amount of fresh water which is an obstacle in south part of the country such as Shiraz city. Therefore this study has been one of the first attempts to thoroughly examine the state of the art aerobic pile composting using municipal market and garden waste in an established compost facility in a mega city in Iran.
Point 2: What is the main objective of the work?
Response 2: The main objective of this study was to introduce the possibility of windrow composting for market and garden waste in large scale for mega cities with established compost facility in Iran. It is tried to explain this at line 83-85 (final version) as followed:
“This study aims to adopt an effective waste treatment model to convert source-separated organic fractions into value-added products under existing SWM system in Iran. The main objective of this work is to practice windrow composting for segregated market and garden wastes as a viable technique in mega cities with established compost facility. The feasibility of suggested system is mainly due to its low initial investment, simple technology and routine monitoring.”
Point 3: The separation of the garbage was carried out directly at the source, an interesting idea on a small scale, but on a large scale is this possible? Is there feasibility for such a study?
Response 3: In mega cities of Iran, business units that generate >5 kg organic waste must contract with municipality to receive day to day collection services. In Shiraz, the 5th populated city (1.6 million populations), 30% of them are given two storage bins with different colours for wet/dry waste segregation. Organic waste collected has been subjected to vermicomposting in small scale for more than 6 years. After successful implementation of this project, Shiraz solid waste management organization increased the number of daily markets targeted for this program. It was mainly due to the fact that comparing to vermicomposting (in average 3 months), this method could turn the organic materials in 10 weeks into value-added product. Furthermore, it requires not as much of labour work and the process is less vulnerable to semi-arid weather condition. In 2019, they started to produce approximately 30 tons per month compost with this method.
Point 4: The time of the composting test must be included in the methods, as well as the reasons why it led the authors to determine this working time.
Response 4: The sampling intervals and the reason for choosing them are inserted in methodology at line 158-164 (final version). Consequently, table 3 is removed. The modified paragraph is as followed:
“The sampling intervals were varied to determine the effect of different parameters. Since moisture content significantly affects the biological activities, it was measured weekly. EC, pH, total N, carbon and density were measured every other week. This time interval was chosen as taking more samples (every week) could inhibit the composting process. Phosphorus (P) and potassium (K) have a lower decomposition rate, therefore their evolution was measured at the beginning and at the end of the process. Parameters that are more crucial to agricultural applications (heavy metals, pathogens, phytotoxicity) were measured only at the end of the process.”
Point 5: As the authors can confirm an assay if only one assay was performed with each sample. Couldn't there have been errors regarding the operator or equipment?
Response 5: Concerning the lack of replicates, the histograms were drawn by using the average of values. Indeed, laboratories analysis was carried out on three subsamples from main sample. It is inserted at line 169-170 in methodology (final version).
The boxplot are used for EC, pH and bulk density-volatile solids are to indicate min, max, median and average. However, C/N ratio figure still shows the average value of all samples.
Point 6: Even without the production of leachate, contaminants, coliforms, protozoa and others that may be present in the material for composting, which may have generated soil pollution. As they ensured that there was no contamination.
Response 6: The point mentioned by Reviewer is completely right. However, according to Iran’s compost standard only E.coli and Salmonella should be checked in organic fertilizers before agricultural applications. Therefore, in this study only these two parameters were considered.
Point 7: With regard to pathogens, the analyses were performed only at the end of the trial, how to ensure that the process was efficient for the treatment since you do not know how it is at the beginning?
Response 7: The pathogen content was measured to ensure the final product quality before agricultural applications. Since it cannot surly hint that the process deactivates the pathogens the sentence is re-phrased as: “The results, presented in Table 4, indicate a pathogen free final product which could be safely applied for cultivation of any crops.”
Point 8: In the conclusions between lines 468 and 474, were the results obtained in the study? Or were they based on bibliographic research? If it is based on the tests, which tests did you perform? How was it proven? A conclusion must be made based on the studies; other statements must be inserted in the text and duly referenced.
Response 8: The increasing of Soil Organic carbon was estimated based on literature. Therefore it is kept as part of discussion to project the benefit of applying compost to soil and is removed from conclusion. The last paragraph of the conclusion is modified by explaining the significance of the findings and contribution of the study as followed:
“The findings of this study suggest that a wet/dry source- separation program can be effectively adopted by business units in Iran. Indeed, in the initial phase, among all municipal organic waste streams, segregation and the separate collection of market and garden waste are more viable under the existing SWM system. These findings are particularly relevant to decision makers in mega cities with established compost facilities”

Round 2
Reviewer 2 Report
All comments have been addressed.
Author Response
The manuscript was spell checked.
Reviewer 3 Report
Dear Authors,
as already reported in first revision I found that the focus of the manuscript is of actual interest. Some highlited "confusing" points related to the experimental approach and Tables are now clear, the manuscript improved.
Best regards
Author Response
The manuscript was spell-checked.
Reviewer 4 Report
Answers 1, 3 and 6 are well-founded. However, this should be included in the manuscript for better explanations regarding the understanding of the chosen methods.
Author Response
Point 1: inserting answer 1 in manuscript
Response 1: the purpose of the work was explained in introduction, however the point regarding shortage of fresh water was added: Line of 81-83 (final version)
Point 2: inserting answer 3 in manuscript
Response 2: The feasibility of the study inserted in conclusion to describe that this method is viable under existing situation: Line 468-472 (final version)
Point 3: inserting answer 6 in manuscript
Response 3: It was mentioned that Iran national standard defines limits only for E.coli and Salmonella content of organic fertilizers: Line 355 (final version)